# Age and Visual Contribution Effects on Postural Control Assessed by Principal Component Analysis of Kinematic Marker Data

**DOI:** 10.3390/sports11050098

**Published:** 2023-05-05

**Authors:** Arunee Promsri

**Affiliations:** 1Department of Physical Therapy, School of Allied Health Sciences, University of Phayao, Phayao 56000, Thailand; arunee.pr@up.ac.th; Tel.: +66-54-466-666-(3817); 2Unit of Excellence in Neuromechanics, School of Allied Health Sciences, University of Phayao, Phayao 56000, Thailand

**Keywords:** postural control, neuromuscular control, balance, movement synergy, aging, unstable surface, closed eye, principal component analysis (PCA)

## Abstract

Postural control, the ability to control the body’s position in space, is considered a critical aspect of health outcomes. This current study aimed to investigate the effects of age and visual contribution on postural control. To this end, principal component analysis (PCA) was applied to extract movement components/synergies (i.e., principal movements, PMs) from kinematic marker data of bipedal balancing on stable and unstable surfaces with eyes closed and open, pooled from 17 older adults (67.8 ± 6.6 years) and 17 young adults (26.6 ± 3.3 years), one PCA-analysis for each surface condition. Then, three PCA-based variables were computed for each PM: the relative explained variance of PM-position (PP_rVAR) and of PM-acceleration (PA_rVAR) for measuring the composition of postural movements and of postural accelerations, respectively, and the root mean square of PM-acceleration (PA_RMS) for measuring the magnitude of neuromuscular control. The results show the age and visual contribution effects observed in PM_1_, resembling the anteroposterior ankle sway in both surface conditions. Specifically, only the greater PA_1__rVAR and PA_1__RMS are observed in older adults (*p* ≤ 0.004) and in closed-eye conditions (*p* < 0.001), reflecting their greater need for neuromuscular control of PM_1_ than in young adults and in open-eye conditions.

## 1. Introduction

Postural control is one of the most critical aspects of health and the ability to accomplish daily motor tasks. The postural control system regulates the body’s position in space to control orientation and stability, which requires the practical cooperation of sensorimotor functions and muscle strength in contributing to joint stabilization [1]. Clinically, the ability of postural control is often considered when assessing the progress of fall prevention programs, typically through measuring the posturography of the body sway, e.g., by assessing the center-of-pressure (COP) displacements [2,3]. However, posturographic information obtained on this basis reflects the outcome of the ground reaction force and moment, indirectly measuring human postural control via COP-based variables (e.g., sway area, sway path length, and sway velocity) [4]. In other words, measuring posturography may provide insufficient information about neuromuscular control in terms of which postural muscles are involved in generating postural sway (i.e., driving COP motions) [5]. In this sense, the possible mechanisms for controlling human posture have been suggested to be studied by directly analyzing movements and muscle activations [5].

Regarding movement analysis, the cooperative contribution of the multiple body segments is necessary to complete the given motor tasks [6,7,8]. However, in order to eliminate the inherent redundancy of the motor apparatus [9], the central nervous system (CNS) is believed to find a near-optimal solution to control human movement while modularly performing a goal-directed motor task through task-relevant synergistic muscle activations [10,11]. Like muscle synergies, motor activities (i.e., motor behaviors) have also been revealed as a combination of task-dependent movement synergies formed together to achieve the given task goal [12,13], typically decomposed by applying the kinematic principal component analysis (PCA), one of the dimensionality-reduction methods, to postural movements [7,14,15,16,17]. Postural movement refers to changes in the body’s posture or the changes in the relative positions and orientations of the body segments [6]. The movement components resulting from PCA called “principal movements” (PM_k_; k denotes the order of movement components) are one-dimensional different movement components that form together to achieve the given task goal. For example, in order to maintain stability for the balance tasks, although the anteroposterior and mediolateral ankle strategies are the first two main principal movements observed for unipedal [7,18] and bipedal [6,7] stances on a rigid surface, other movement strategies (e.g., hip strategies) have been observed [7]. As previously reported, applying a PCA to kinematic data allows assessing the coordinative structure and the control of individual postural movement components (i.e., how the postural control system structures and controls the movement of its segments) [6]. The coordinative structure (i.e., composition) of postural movements is represented by the amount of activity of the individual PMs. Differences in the relative contribution of the PMs show that the interplay of PMs and hence the coordinative structure of this movement are different [18,19]. In addition, the control of individual movement components can be analyzed using the PM accelerations since the PM accelerations are a direct result of the control system, as accelerations are directly proportional to the acting forces mainly caused by muscle acceleration [20] directly allowed for measuring the neuromuscular control [6,17,20]. 

Although quiet standing is one of the motor-balance skills of everyday life, maintaining postural balance requires not only a healthy musculoskeletal system related to muscle strength [21] but also the automatic regulation of the CNS by the central integration of vestibular, visual, proprioceptive, and tactile information [22]. Since movement strategies can be flexibly adapted to meet internal (e.g., age effects [23,24] and visual contribution [24,25]) and external (e.g., altered support surface [6,26]) demands, this current study aimed to better understand the effect of two factors—age and visual contribution—on bipedal postural control by focusing on the inherent ability of the postural control system in structuring and controlling bipedal postural movements. With advancing age, age-related changes in the neural, sensory, and musculoskeletal systems can lead to balance impairments that significantly impact the ability to move about safely [27]. Specifically, proprioception (i.e., sense of body position and movement) encompasses signals from mechanoreceptors (proprioceptors) located in muscles, tendons, and joint capsules, especially from leg muscles, which provide the primary source of information for postural control [28]. Alterations of the proprioceptive signal according to advancing age in the legs and the present compelling evidence have been reported as changes modifying the neural control of upright standing by inducing a decrease in the sensitivity, acuity, and integration of the proprioceptive signal [29]. Moreover, since vision is one of the basic sensory systems regulating postural control [22], vision deprivation should affect an ability to maintain balance. More challenging postural control tasks can be created by altering those necessary sensory inputs, e.g., by performing the balancing task with the eye closed. 

In order to analyze postural control, PCA is widely applied to postural movements or kinematic marker data to identify movement synergies that work together to achieve a given task goal, e.g., maintaining balance while standing [23,24,30,31,32,33]. Movement synergies refer to groups of muscles (e.g., muscle synergies) [34] or movement patterns (e.g., movement components) [7] coordinated to achieve a specific movement goal. Therefore, applying PCA to kinematic postural movements is one alternative approach to better understand which movement components/synergies cooperate to complete the balance task goal [35]. For example, visualization of the movement components shows different movement components/synergies contribute to maintaining balance, e.g., the ankle or hip sway or strategy [7,17]. This feature lets researchers see the main movement synergies to achieve the task goal. Moreover, this method has been applied to determine the underlying mechanisms of postural control and how they are affected by different factors, e.g., injury risk factors [26,36], feet support area [6], or disease/syndrome [33]. In this sense, this current study focused on these movement components/synergies that can be analyzed to understand the effects of aging or changes in visual feedback on postural control. Therefore, investigating the underlying movement components/synergies contributing to equilibrium can provide insights into how these factors affect postural control, specifically in terms of multiple body segments cooperatively working to maintain balance. 

In summary, this current study aimed to investigate the effects of age and visual contribution on postural control by analyzing individual movement components/synergies (i.e., principal movements, PMs) that contribute to bipedal equilibrium. PCA was used to extract the PMs, focusing on the composition and control of bipedal postural movements while bipedal balancing on stable and unstable surfaces under two visual conditions: closed and open eyes. As previously reported, advancing age [23] and omitted vision [24,25] can alter specific movement components of the postural control system rather than affecting the system as a whole. Thus, it was hypothesized that the effects of age and visual contribution would be observed in specific movement components relevant to the current tasks. Furthermore, understanding inherent bipedal postural control can also benefit neuromuscular control training and injury prevention, such as falls in elderly adults. 

## 2. Materials and Methods

### 2.1. Secondary Data Analysis

The kinematic datasets of 34 participants standing under different conditions used in this current study were obtained from a peer-reviewed open-access dataset [37]. The local ethics committee of the Federal University of ABC (CAAE: 53063315.7.0000.5594) approved the study protocol, and all participants signed consent before participating. The original datasets consist of 49 participants’ kinematics data. However, fifteen participants were not included in this current study due to the presence of health problems without correction (e.g., cerebral palsy [38], n = 1, excessive body weight (BMI > 30) [39], n = 10; labyrinthitis [40], n = 1; scoliosis [41], n = 1) that might influence the postural control ability, leading to only the kinematics dataset of 34 participants (17 young adults [9 males and 8 females] and 17 older adults [8 males and 9 females]) being included for further analysis. The participant characteristics are represented in Table 1. 

Measurement procedures were fully described by dos Santos et al. [37]. In brief, before performing the experiments, each participant was equipped with 42 reflective markers placed on the anatomical landmarks based on the marker placement and segment definition proposed by Leardini et al. [42,43]. However, the markers placed on the upper limbs were not tracked because participants were instructed to maintain the placement of the arms along the trunks during the trials. A motion capture system consisting of 12 infrared cameras with a sampling frequency of 100 Hz (Raptor-4, Motion Analysis, Santa Rosa, CA, USA) was used to record the full-body 3D kinematics of each participant during the quiet standing trials, which was operated through the Cortex software version 5.3 (Motion Analysis, Santa Rosa, CA, USA). The kinematic data were filtered with a 10 Hz, 4th-order zero-lag low-pass Butterworth filter. 

Regarding the experimental protocol, the participants were evaluated barefoot, standing still for 60 s on stable and unstable surfaces with two eye conditions: eyes open and closed. The order of the balancing conditions was randomized for each participant. For all the balancing trials, all participants were asked to place their feet at an angle of 20 degrees and their heels 10 cm apart over the lines marked on the stable surface and the balance pad [4]. For unstable conditions, two balance pads (Airex AG, Sins, Switzerland) were used, one for each foot. During testing, the participants were required to stand barefoot and as still as possible with their arms at their sides in all conditions. In the open-eye conditions, each participant was instructed to look at a 5 cm round black target placed at the individual’s eye level on a wall in front of the participant. In closed-eye conditions, the participants were first asked to look at the target with their eyes open, regulate to find a stable and comfortable posture given the requirements, and then close their eyes. 

Although the assessments were repeated three times per balancing condition, this current study included only one trial of each balancing condition with no missing markers and no incomplete recording problems for further analysis. This checking process was performed by running each of the C3D files.

### 2.2. Movement Synergy Extraction

All data processing was conducted in MATLAB version 2022a (MathWorks Inc., Natick, MA, USA). Typically, PCA is a statistical technique that transforms a dataset with many variables into a smaller set of new variables called principal components (PCs), which capture the most critical information in the original data used for data reduction, visualization, and exploratory data analysis [15,44]. It identifies data patterns and compresses them into smaller variables that explain most of the variance in the original dataset [15,44]. Each PC is a linear combination of the original variables, with the property that the first component explains the largest possible variance, and each subsequent component explains the largest possible variance that is orthogonal (uncorrelated) to the previous components [15,44]. When PCA is applied to this kinematic marker data by identifying the most important patterns or “components” of movement, they explain the majority of the variance in the data [35]. These components are essentially linear combinations of the original variables, and they can be thought of as different movement components/synergies called principal movements (PMs) that coordinate to complete the given task goal [35].

In this current study, PCA through the PManalyzer software [39] was applied to extract the PMs contributing to bipedal equilibrium from the kinematic marker data of bipedal balancing on stable and unstable surfaces with closed and open eyes. Two PCA analyses were computed for each support surface condition to avoid the support surface movement influencing the postural control movementssupport surface movement influencing the postural control movements [6,26], as seen in Appendix A, respectively. In each PCA analysis, an individual dataset contained the kinematic data of 42 markers contributing 126 spatial coordinates (x, y, z) interpreted as 126-dimensional posture vectors. Then, three pre-processing analyses were carried out for each dataset: (I) centered by subtracting the mean posture vector to prevent differences in mean marker positioning in space from influencing the PCA outcome [35], (II) normalized to the mean Euclidean distance to address anthropometric differences [35], and (III) weighted by considering sex-specific mass distributions [45], of which the mathematically detailed procedures were fully described by previous studies [35,36,46]. Then, the weighted postural vectors from all volunteers were concatenated to form a 408,000 × 126 input matrix (100 [sampling rate] × 2 [number of trials (closed-eye and open-eye conditions)] × 60 [testing duration (s)] × 34 [number of participants] × 126 [marker coordinates]) for further PCA. 

PCA was performed using a singular-value decomposition of the covariance matrix to decompose all kinematic data into a set of orthogonal eigenvectors, i.e., principal components (PC); k denotes the order of movement component. Animated stick figures can be created to characterize each eigenvector’s movement pattern (i.e., PM_k_) [35]. The actual time evolution (time series) of individual PM_k_ is quantified by the PC scores or “principal positions” (PP_k_(*t*)), which represent positions in posture space, i.e., the vector space spanned by the PC-eigenvectors [35]. The word ‘’principal” in the variable names denotes that these variables were obtained through a PCA, and (*t*) indicates that these variables are functions of time *t* [35]. Regarding Newton’s mechanics, the second-time derivatives, “PM_k_-accelerations or principal accelerations” (PA_k_(*t*)), can be calculated from the PP_k_(*t*) according to the conventional differentiation rules [35]. The associations between PA_k_(*t*) and myoelectric activity were demonstrated for postural control tasks [20], supporting that PA_k_-based variables can reveal the characteristics of neuromuscular control of individual PM_k_ [6,18,23,35,36,46,47]. In order to avoid noise amplification in the differentiation processes, a Fourier analysis was conducted on the raw PP_k_(t) [46], revealing that the highest power resided in frequencies around 2–5 Hz, but that visible power was still observed in the frequency range between 6 and 10 Hz. Therefore, the time series were filtered with a 3rd-order zero-phase 10 Hz low-pass Butterworth filter before performing the differentiation step.

This current study used leave-one-out cross-validation to evaluate the vulnerability of the PM_k_ and the dependent variables to changes in the input data matrix to address validity considerations [35]. Therefore, the first five PCs of balancing on each surface condition (stable and unstable surfaces) proved robust and were selected to test the hypotheses. In addition, the cumulative eigenvalues of the PC_1–5_ or the relative explained variance of PP_k_ used in this current study of both balancing conditions were higher than 90%, which reached the standard criterion for selecting the movement components (i.e., PCs) that could account for most of the variance within the data [48]. 

### 2.3. PCA-Based Variable Computation

In order to determine the coordinative structure or composition of postural movements, the subject-specific *relative explained variance* (rVAR) of PP_k_ or PP_k__rVAR was calculated from the PP_k_(t). The PP_k__rVAR quantifies how much (in percent) each PM contributed to the total variance in postural positions [18,20,26,46,49,50]. Differences in PP_k__rVAR between conditions indicate a difference in the coordinative structure of postural movements, i.e., a different contribution of individual PM_k_ to total postural variances. 

In order to investigate the control of individual movement components, two subject-specific PA_k_-based variables were computed. First, the participant-specific *relative explained variance* (rVAR) of the PA_k_(t) or PA_k__rVAR was computed to quantify how much (in percent) individual PM_k_-accelerations contributed to the total variance in postural accelerations [26,46,49,51]. In other words, these PA_k_-based variables reflect how fast the individual movement components change and are accelerated [46,51]. Differences in PA_k__rVAR between conditions indicate a difference in the contribution of individual PM_k_ acceleration to total postural acceleration variances. Second, the *root mean square* (RMS) of PA_k_(*t*) or PA_k__RMS was calculated as a measure of the magnitude of individual PM_k_ acceleration [47,52], of which differences in PA_k__RMS between conditions indicate different magnitudes of neuromuscular control [47,52].

### 2.4. Statistical Analysis

All statistical analyses were performed using the IBM SPSS Statistics software version 26.0 (SPSS Inc., Chicago, IL, USA), with the alpha level set at a = 0.05. Shapiro–Wilk tests suggested a split-plot repeated measures ANOVA for testing the main and interaction effects of age (between subjects) and visual contribution (within subjects) for each PCA-based variable (PP_k__rVAR, PA_k__rVAR, and PA_k__RMS). In addition, the effect size (Partial Eta Squared value, η_p_^2^) and the observed power (1 − *β*) were reported. 

## 3. Results

### 3.1. Movement Components

Table 2 represents descriptive movement characteristics of the first five movement components (PM_1–5_). For stable surface conditions, the first eight PMs together explain 91.4% of the total relative variance of postural positions and 17.6% of the total relative variance of postural accelerations. For the unstable surface conditions, the first eight PMs together explain 97.3% of the total relative variance of postural positions and 18.6% of the total relative variance of postural accelerations. 

Furthermore, video representations of PM_1–5_ are shown in Appendix A for the stable condition and Appendix A for the unstable condition, respectively, with both visualizations amplified 2X for clarity. The anteroposterior ankle strategy is the first main movement component (PM_1_) to achieve bipedal balancing on stable and unstable surfaces.

### 3.2. Age Effects

For stable conditions (Figure 1; left column), only the age effects on controlling individual movement components are observed in the specific PMs. Specifically, older adults have a greater contribution of postural acceleration in PM_1_ (PA_1__rVAR; *p* = 0.004, η_p_^2^ = 0.227, 1 − *β* = 0.844), but have a smaller contribution of postural acceleration in PM_5_ (PA_5__rVAR; *p* = 0.001, η_p_^2^ = 0.309, 1 − *β* = 0.956) than young adults.

For unstable conditions (Figure 1; right column), the age effect in the composition of bipedal postural movements is observed in PM_3_ (PP_3__rVAR; *p* = 0.005, η_p_^2^ = 0.220, 1 − *β* = 0.830), of which a smaller contribution in this movement component found in older adults than young adults. Moreover, in PM_1_, older adults also show a greater contribution of postural accelerations (PA_1__rVAR; *p* < 0.001, η_p_^2^ = 0.410, 1 − *β* = 0.995) and the magnitude of neuromuscular control (PA_1__RMS; *p* = 0.004, η_p_^2^ = 0.228, 1 − *β* = 0.847) than young adults. 

### 3.3. Visual Contribution Effects

For stable conditions (Figure 2; left column), the effect of visual contribution is only observed in the control of individual movement components, of which a greater proportion of postural acceleration in PM_1_ (PA_3__rVAR; *p* = 0.002, η_p_^2^ = 0.272, 1 − *β* = 0.918) observed in the closed eye condition than the open eye condition.

For unstable conditions (Figure 2; right column), only the visual contribution effects are only observed in the specific PMs, of which a greater contribution of postural acceleration in PM_1_ (PA_1__rVAR; *p* < 0.001, η_p_^2^ = 0.431, 1 − *β* = 0.998) and in PM_3_ (PA_3__rVAR; *p* = 0.009, η_p_^2^ = 0.196, 1 − *β* = 0.772) found in the closed eye condition than the open eye condition. Moreover, the closed eye conditions also show greater magnitudes of neuromuscular control than the open eye condition in PM_1_ (PA_1__ RMS; *p* < 0.001, η_p_^2^ = 0.575, 1 − *β* = 1.000), PM_2_ (PA_2__ RMS; *p* = 0.005, η_p_^2^ = 0.218, 1 − *β* = 0.825), PM_3_ (PA_3__ RMS; *p* = 0.002, η_p_^2^ = 0.258, 1 − *β* = 0.899), PM_4_ (PA_4__ RMS; *p* < 0.001, η_p_^2^ = 0.356, 1 − *β* = 0.983), and PM_5_ (PA_5__ RMS; *p* < 0.001, η_p_^2^ = 0.430, 1 − *β* = 0.997). 

### 3.4. Interaction Effect

The interaction effect between the age and visual contribution is only observed in PM_2_ (PA_2__rVAR; *p* = 0.001, η_p_^2^ = 0.276, 1 − *β* = 0.923) for unstable conditions.

## 4. Discussion

This current study investigated the effects of age and visual contribution on the composition and control of bipedal postural movements. The postural movements during balancing on stable and unstable surfaces were separately extracted into a set of movement components/synergies, i.e., principal movements (PM_k_), through a principal component analysis (PCA). For each surface condition, two types of PCA-based variables based on PP_k_ and PA_k_ were computed as measures of the coordinative structure of postural movements (PP_k__rVAR) and the control of individual movement components (PA_k__rVAR and PA_k__RMS) in terms of quantifying the composition of postural acceleration and the magnitude of neuromuscular control, respectively. As expected, the main findings show that the effects of age and visual contribution emerge in the specific movement component. 

Regarding advancing age, older and young adults use the same movement synergy (PP_1__rVAR), which resembles the anteroposterior ankle sway, to achieve bipedal balancing on stable and unstable surfaces. However, the higher contributions of postural accelerations (PA_1__rVAR) of this movement component, reflecting how fast this postural movement changes and how much it is accelerated [46], are observed for older adults and are seen in both surface conditions. These findings indicate that the postural sway of older adults in this movement component was faster than that of young adults, reflecting an increased instability of the anteroposterior ankle sways with increasing age. This interpretation is supported by the greater magnitude of neuromuscular control (PA_1__RMS) found in older adults than in young adults when balancing with unstable surface conditions. Specifically, older adults have faster postural sway and decreased stability control than young adults, particularly on unstable surfaces seen in the first main movement component, the anteroposterior ankle strategy. Together, these findings suggest that the ability of the neuromuscular system to control stability during an upright stance is inherently diminished with advancing age, especially when balancing on unstable surfaces. 

Like the age effects, the exact main movement synergy observed when performing bipedal balancing on stable and unstable surfaces under closed and open eye conditions resembled the anteroposterior ankle sway (PM_1_). However, only differences in the control of movement components were observed in the main movement components. In particular, balancing with closed eyes results in higher postural accelerations (PA_1__rVAR) of this movement component than balancing with open eyes in both surface conditions, indicating faster anteroposterior ankle sways during closed-eye balancing [46]. In other words, closing eyes during balancing results in faster postural sway and decreased stability control of the first main movement component, the anteroposterior ankle sway, especially on unstable surfaces, compared to balancing with open eyes. These findings support the hypothesis that balancing with the closed eye condition makes it more challenging to control postural stability than balancing with open eyes, mainly when the balance tasks are performed on unstable surfaces since a high magnitude of neuromuscular control (PA_1__RMS) is observed in unstable conditions for the closed eye condition. 

When focusing on the first main movement component (PM_1_), the anteroposterior ankle strategy (PM_1_) is used to achieve bipedal balancing on both stable and unstable surfaces, as previously reported [6,7,17], and reflects an inverted pendulum model [53]. According to these observations, the neuromuscular control of the muscle groups (e.g., ventral and dorsal muscles of the ankle, knee, and hip joints) that play an essential role in flexibly stabilizing the upright posture in the anteroposterior ankle strategy may be beneficial for maintaining and gaining ability in postural control, especially for older adults in preventing falls. Impaired postural control leads to falls [54], in which fallers sway more in a quiet or perturbed stance than non-fallers [55]. In addition, closed-eye conditions could be applied to postural control training since they facilitate the neuromuscular functions of the whole-body segments in cooperatively achieving equilibrium.

For clinical applications, the main empirical findings suggest that the inherent age and visual contribution effects on postural control should be considered for training, injury prevention, and rehabilitation. For example, higher neuromuscular control of the first main movement strategy (the anteroposterior ankle sway seen in PM_1_) in older adults and in balancing with closed eyes reflects the inferior neuromuscular performance caused by inherent degenerative changes in the neuromuscular system with advancing age [56] and by omitting visual input [22]. Specifically, the dorsi flexor and plantar flexor muscles play an essential role in the anteroposterior ankle strategy [20], and their age-related degenerations have been reported as a possible risk of falls [57]. Moreover, since age-related changes in the neuromuscular system and omitting visual input can impair postural control and increase the risk of falls and injuries [58], it is essential to consider these factors to prevent injuries and promote recovery from injury. Older adults may benefit from specific training programs that focus on enhancing neuromuscular control of not only the lower extremities but also other body parts (e.g., core [59] and upper limb [60] muscles) to slow the effects of aging on postural control since the neuromuscular system controls posture through multiple muscles in producing relative movements between body segments [5,6,20]. Similarly, individuals who have experienced an injury affecting postural stability may need tailored rehabilitation protocols that target neuromuscular performance, of which balancing training on unstable surfaces and omitting visual feedback is suggested to improve their ability to maintain postural stability [61]. 

### Limitations and Future Study

One limitation of this current study is that there is no marker tracking of the upper extremity, and the participants are instructed to keep their arms along the body site, as reported in the original data article [37]. Nevertheless, although only a single trial per condition from each participant was analyzed, the quality of the selected signal or dataset was good with no missing data, and all the participants could complete the balance tasks without any situations affecting maintaining equilibrium, e.g., falling or stepping out of the starting position. 

For future studies, considering whole-body segment trajectories is suggested for studying postural control, since all body segments cooperatively generate movements to control postural stability. In addition, applying other dimensional reduction methods [62] or conducting a combination study between the dimensional reduction method and electromyographic [20] or center-of-pressure (COP) [25,32] analysis to analyze postural control may be of interest, possibly providing other relevant information about maintaining equilibrium.

## 5. Conclusions

The current findings highlight the effective extraction of the movement components/synergies (i.e., principal movements; PM_k_) from the whole-body bipedal postural movements through a principal component analysis (PCA) in revealing the effects of age and the visual contribution seen in the specific PMs. Specifically, older adults have higher control of the anteroposterior ankle sway (PM_1_) than young adults. In addition, balancing with closed eyes also shows higher control of this movement component than balancing with open eyes. Therefore, knowledge of the inherent movement strategies used to achieve equilibrium should be considered for postural control training.

## Figures and Tables

**Figure 1 sports-11-00098-f001:**
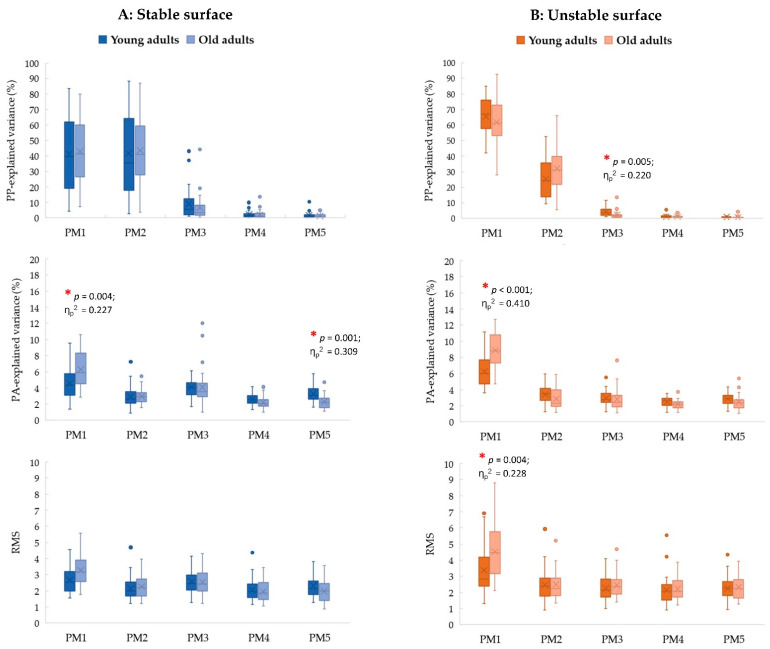
The post hoc comparisons of the *relative explained variance* of PP_k_ (PP_k__rVAR; first row), the *relative explained variance* of PA_k_ (PA_k__rVAR; second row), and the *root mean square* (RMS) of PA_k_ (PA_k__RMS; third row) between young and older adults separated by each support surface: (**A**) stable surface (**left** column) and (**B**) unstable surface (**right** column) (* *p* < 0.01).

**Figure 2 sports-11-00098-f002:**
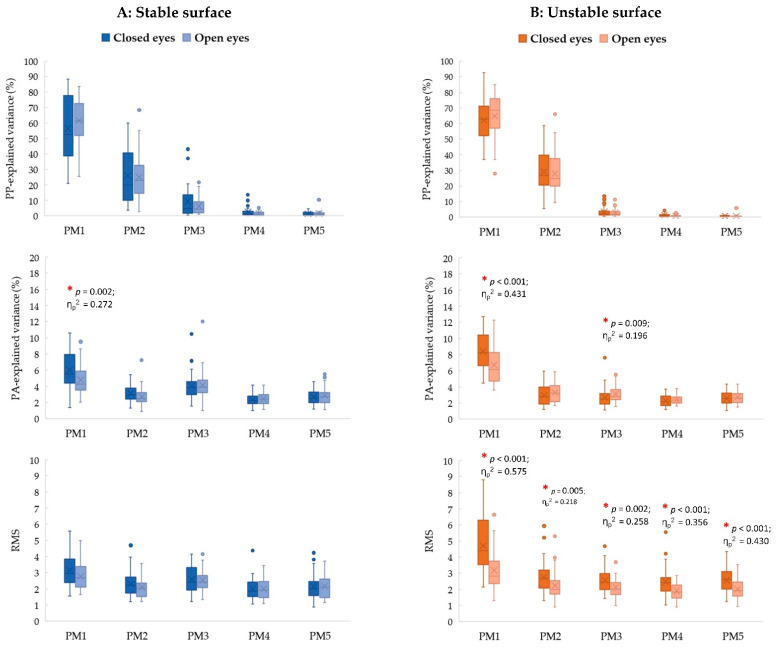
The post hoc comparisons of the *relative explained variance* of PP_k_ (PP_k__rVAR; first row), the *relative explained variance* of PA_k_ (PA_k__rVAR; second row), and the *root mean square* (RMS) of PA_k_ (PA_k__RMS; third row) between closed and open eye conditions separated by the support surfaces: (**A**) stable surface (**left** column) and (**B**) unstable surface (**right** column) (* *p* < 0.01).

**Table 1 sports-11-00098-t001:** Characteristics of participants (Mean ± SD; * *p* < 0.05).

	Young (n = 17)	Older (n = 17)	*p*-Value
Age (yrs.)	26.6 ± 3.3	67.8 ± 6.6	<0.001 *
Weight (kg)	66.3 ± 12.3	65.4 ± 8.3	0.807
Height (cm)	172.2 ± 1.1	161.3 ± 0.8	0.002 *
Body mass index (kg/m^2^)	22.3 ± 3.3	25.1 ± 2.3	0.007 *

**Table 2 sports-11-00098-t002:** The relative explained variance of principal positions (PP_k__rVAR (%)) and principal accelerations (PA_k__rVAR (%)), and the main descriptive movements of the first five principal movements (PM_1–5_) separately analyzed each support surface: (**A**) a stable surface and (**B**) an unstable surface. Note: the closed-eye and open-eye balancing trials were pooled and analyzed for each surface condition.

PM_k_	PP_k__rVAR	PP_k__rVAR	Descriptive Characteristics
**A: Stable surface**			
1	59.1 ± 17.8	5.4 ± 2.2	Anteroposterior sway around the ankle joint
2	25.5 ± 15.5	2.9 ± 1.1	Mediolateral sway around the ankle joint
3	7.9 ± 9.2	4.1 ± 1.8	Anteroposterior sway around the hip joint
4	2.1 ± 2.5	2.4 ± 0.7	Whole-body rotation
5	1.5 ± 1.7	2.8 ± 1.0	Head flexion and extension combined with the hip rotation
**B: Unstable surface**			
1	63.6 ± 13.0	7.6 ± 2.4	Anteroposterior sway around the ankle joint combined with the support surface movement
2	28.6 ± 12.5	3.1 ± 1.2	Mediolateral sway around the ankle joint
3	3.2 ± 2.8	2.9 ± 1.1	Anteroposterior sway around the ankle joint combined with anteroposterior sway around the hip joint
4	1.1 ± 1.1	2.3 ± 0.6	Whole-body rotation
5	0.8 ± 1.0	2.7 ± 0.9	Whole-body rotation combined with knee flexion and extension

## Data Availability

The raw kinematic marker data are available in dos Santos et al. [37]. For the PCA-based time series data, interested parties may request them from the author.

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
