# Peer review of "Age and Visual Contribution Effects on Postural Control Assessed by Principal Component Analysis of Kinematic Marker Data"

_sports, 2023, doi:10.3390/sports11050098_

Round 1

Reviewer 1 Report

The paper focuses on the effects of age and vision on postural control, proposing a statistical analysis based on non-supervised machine learning techniques. The paper is well-written and clear. Nonetheless, there are some concerns and notes: 1) The title might be improved to better stress the fact that the paper actually performs a data analysis 2) the abstract is not super well organised and might be improved. Maybe all the acronyms can be deleted 3) The methodology is clear, but the authors should clarify how and why they selected those features for their analysis 4) the authors wrote that the data availability statement does not apply. However, imho it would be interesting having access to the collected dataset

Author Response

Response to Reviewer 1 Comments

Point 1: The paper focuses on the effects of age and vision on postural control, proposing a statistical analysis based on non-supervised machine learning techniques. The paper is well-written and clear. Nonetheless, there are some concerns and notes: 1) The title might be improved to better stress the fact that the paper actually performs a data analysis

Response 1: Thank you for taking the time to review the article. I appreciate your insightful comments and suggestions. Your feedback has been precious in improving the quality of the work. I have carefully considered your suggestions and made the necessary revisions to address your issues.

According to the suggestion for the manuscript title, I have changed the title to "Age and Visual Contribution Effects on Postural Control Assessed by Principal Component Analysis of Kinematic Marker Data."

Point 2: 2) the abstract is not super well organised and might be improved. Maybe all the acronyms can be deleted

Response 2: Thank you very much. I slightly rewrote and rearranged the abstract's contents, hoping it is clearer than the old manuscript.

Point 3: 3) The methodology is clear, but the authors should clarify how and why they selected those features for their analysis

Response 3: Thank you very much. I added another paragraph explaining the purpose, method, and tested factors.

Point 4: 4) the authors wrote that the data availability statement does not apply. However, imho it would be interesting having access to the collected dataset

Response 4: Thank you for your suggestion. I add the data information and provide the PCA-based data on request.

Reviewer 2 Report

Comments to Authors

1 – At the end of section 1, please add a paragraph stating the organization of the remainder of the paper.

2 – End of section 2.2. Line 178. We have “The first five PCs proved robust “ The author can provide more details on the criteria used here to select these five PC.

3 – The author may provide more details on the motivation to choose PCA. Is it the best approach on previous studies? In the literature or on the authors own work, are there any other techniques suitable for this purpose?

4 – A suggestion: on the future work part, the author may state that a direction for future work id to use another dimensionality reduction different from PCA.

Comments on Writing

Line 50

Please change “segments)” -> “segments”

Line 115

Please correct “Santos et al. [35].” The first author name in [35] is not Santos.

Line 194

Please change “the root mean square of” -> “the root mean square (RMS) of”

Please define the acronym on its first occurrence.

Ref. 38 seems to be incomplete. Please check.

Author Response

Response to Reviewer 2 Comments

Point 1: Comments to Authors

1 – At the end of section 1, please add a paragraph stating the organization of the remainder of the paper.

Response 1: Thank you for taking the time to review the article. I appreciate your insightful comments and suggestions. Your feedback has been precious in improving the quality of the work. I have carefully considered your suggestions and made the necessary revisions to address your issues.

According to your point, I rewrote the end paragraph and hoped it would state the information for the remainder of the paper.

Point 2: 2 – End of section 2.2. Line 178. We have "The first five PCs proved robust "The author can provide more details on the criteria used here to select these five PC.

Response 2: Thank you for the comments. I added more information about the criteria used to select the number of PMs for testing the effects of age and visual contribution.

Point 3: 3 – The author may provide more details on the motivation to choose PCA. Is it the best approach on previous studies? In the literature or on the authors own work, are there any other techniques suitable for this purpose?

Response 3: Thank you for the suggestion. PCA is one dimension reduction method widely applied to kinematic marker data to identify movement components/synergies. This method is suitable for extracting different movement synergies forming together to achieve the balance task goal (i.e., equilibrium). Based on my experiences, PCA is suitable for kinematic marker data. However, other methods, e.g., the non-negative matrix factorization, are suitable for EMG data.

Point 4: 4 – A suggestion: on the future work part, the author may state that a direction for future work id to use another dimensionality reduction different from PCA.

Response 4: Thank you very much for the suggestion. I added your suggestion in the future research study..

Point 5: Comments on Writing

Line 50

Please change "segments)" -> "segments"

Line 115

Please correct "Santos et al. [35]." The first author name in [35] is not Santos.

Line 194

Please change "the root mean square of" -> "the root mean square (RMS) of"

Please define the acronym on its first occurrence.

Ref. 38 seems to be incomplete. Please check.

Response 5: Thank you very much for pointing out my mistakes.

I corrected all of my mistakes already, and also I checked and corrected the reference already.

Reviewer 3 Report

The impacts of age and vision on postural control have been extensively studied. A unique contribution of this work is that it introduces a PCA-based approach to extract features that characterize aging and vision effects. The proposed approach is legitimate and effective. However, the contributions are somehow limited since many features have been proposed for the same bipedal balancing problem. At the same time, it is difficult to claim that the proposed method outperformed the previous methods.

It would be nice if the authors could make some comparative comments about their approach and the conventional force-plate-based approach. This is valuable since the traditional force-based approach is much easier to implement than the proposed approach.

Author Response

Response to Reviewer 3 Comments

Point 1: The impacts of age and vision on postural control have been extensively studied. A unique contribution of this work is that it introduces a PCA-based approach to extract features that characterize aging and vision effects. The proposed approach is legitimate and effective.

However, the contributions are somehow limited since many features have been proposed for the same bipedal balancing problem. At the same time, it is difficult to claim that the proposed method outperformed the previous methods.

It would be nice if the authors could make some comparative comments about their approach and the conventional force-plate-based approach. This is valuable since the traditional force-based approach is much easier to implement than the proposed approach.

Response 1: Thank you for taking the time to review the article. I appreciate your insightful comments and suggestions. Your feedback has been precious in improving the quality of the work. I have carefully considered your suggestions and made the necessary revisions to address your issues.

The current study intended to analyze the effects of age and visual contribution on postural control by focusing on movement components/synergies, providing information on how the body moves.

Assessing postural sway by monitoring COP trajectory is commonly seen in several postural control studies. Hence, using different methods is attractive, especially in providing a new perspective on postural control.

However, combining PCA and COP analyses is exciting and may provide useful information. Therefore, I also added this point in the suggestions for future studies.

Reviewer 4 Report

Review Manuscript ID: Sports-2300588

Note:  I offer only a partial review, as I do not feel qualified to evaluate the application principal component analysis (PCA) in this study.

The study tested the effects of age and visual contribution on postural control by focusing on the composition and control of bipedal postural movements during balancing on stable and unstable surfaces with two visual contributions (closed eyes and open eyes). Overall, while the article is pretty well written, I have concerns, which are detailed below.

First, don’t find this article germane to the journal Sports. There are numerous other journals that are a more appropriate outlet for this research.

Intro:   Even though the introduction is quite lengthy, it is not clear how the current study is novel.  First, the author must more directly state what makes the current study new and informative. Second, the author needs to provide a more compelling (persuasive) rationale for the study.  What is the specific research question being addressed?  Why does the question deserve answering? What is new and important?

L115   Santos et al. [35].  Ref 35 is not Santos et al.

L138-40         Only a single trial per condition from each participant seems dubious.  At a minimum, the author will need to provide a compelling justification for using just one trial.

L129-30         Participants were asked to place their feet at an angle of 20 degrees and their heels 10 cm apart over lines marked on the stable surface and the balance pad.  It was inappropriate to force participants to use what could be unnatural positions and postures for their trial, as this introduced a confounding variable and diminished ecological validity.  Participant position and posture should have been natural, or at least body-scaled, rather than dictated using contrived absolute measures.

L135   How was it verified that participants didn’t open eyes during the closed-eye condition?

L287-88         The author indicates that the findings suggest that the ability of the neuromuscular system to control stability during an upright stance is inherently diminished with advancing age, especially when balancing on unstable surfaces. 

We already knew this.  What’s new here?

L296-299       The author indicates that the findings support the hypothesis that balancing with closed eyes makes it more challenging to control postural stability than balancing with open eyes, mainly when the balance tasks are performed on unstable surfaces…

Again, what’s new here?

L312-13         The author indicates that the findings suggest that the inherent age and visual contribution effects on postural control should be considered for training, injury prevention, and rehabilitation.

Same, what’s new here?

Author Response

Response to Reviewer 4 Comments

Point 1: Review Manuscript ID: Sports-2300588

Note:  I offer only a partial review, as I do not feel qualified to evaluate the application principal component analysis (PCA) in this study.

The study tested the effects of age and visual contribution on postural control by focusing on the composition and control of bipedal postural movements during balancing on stable and unstable surfaces with two visual contributions (closed eyes and open eyes). Overall, while the article is pretty well written, I have concerns, which are detailed below.

First, don't find this article germane to the journal Sports. There are numerous other journals that are a more appropriate outlet for this research.

Intro:   Even though the introduction is quite lengthy, it is not clear how the current study is novel.  First, the author must more directly state what makes the current study new and informative. Second, the author needs to provide a more compelling (persuasive) rationale for the study.  What is the specific research question being addressed?  Why does the question deserve answering? What is new and important?

Response 1: Thank you for taking the time to review the article. I appreciate your insightful comments and suggestions. Your feedback has been precious in improving the quality of the work. I have carefully considered your suggestions and made the necessary revisions to address your issues.

I would like to inform you that the current study was automatically transferred from another journal from MDPI, as suggested by the Editor.

However, I believe that the current study meets the scope of one category of the Sports "Health Professions (Physical Therapy, Sports Therapy, and Rehabilitation."

In the revised manuscript, I added one more paragraph to provide more rationale in the introduction and slightly rewrote the discussions.

Point 2: L115   Santos et al. [35].  Ref 35 is not Santos et al.

Response 2: Thank you very much for pointing out my mistakes. I did check and corrected the references already.

Point 3: L138-40         Only a single trial per condition from each participant seems dubious.  At a minimum, the author will need to provide a compelling justification for using just one trial.

Response 3: Thank you very much for pointing out this point. I added this point to the limitation part of the study. I clarified this point in both Section 2.1 and the study's limitations.

Point 4: L129-30 Participants were asked to place their feet at an angle of 20 degrees and their heels 10 cm apart over lines marked on the stable surface and the balance pad.  It was inappropriate to force participants to use what could be unnatural positions and postures for their trial, as this introduced a confounding variable and diminished ecological validity.  Participant position and posture should have been natural, or at least body-scaled, rather than dictated using contrived absolute measures

Response 4: Thank you very much for sharing valuable information about the starting position.

However, the current study is the secondary data analysis, that I did not design the study protocol. Therefore, your suggestion would be beneficial for my research design in the future.

Point 5: L135   How was it verified that participants didn't open eyes during the closed-eye condition?

Response 5: According to this point, there was no report that participants opened their eyes in the closed-eye balancing conditions. This point can be fixed by using the eye or sleeping pads for future study.

Point 6: L287-88 The author indicates that the findings suggest that the ability of the neuromuscular system to control stability during an upright stance is inherently diminished with advancing age, especially when balancing on unstable surfaces. We already knew this. What's new here?

Response 6: The new findings from the study suggest that older and young adults use the same movement synergy (PP1_rVAR) to achieve bipedal balancing on both stable and unstable surfaces. However, older adults have a higher contribution of postural accelerations (PA1_rVAR) in this movement component, indicating faster and more unstable anteroposterior ankle sways with increasing age. The study also found that older adults had a greater magnitude of neuromuscular control (PA1_RMS) than young adults when balancing on unstable surfaces, suggesting that older adults have a decreased ability to control stability during an upright stance, especially on unstable surfaces. The study suggests that the neuromuscular system's ability to control stability is inherently diminished with advancing age.

Point 7: L296-299 The author indicates that the findings support the hypothesis that balancing with closed eyes makes it more challenging to control postural stability than balancing with open eyes, mainly when the balance tasks are performed on unstable surfaces… Again, what's new here?

Response 7: The new findings suggest that the main movement synergy observed during bipedal balancing on stable and unstable surfaces under closed and open eye conditions is similar to the anteroposterior ankle sway (PM1). However, differences in the control of movement components were observed in the main movement components. Balancing with closed eyes resulted in higher postural accelerations (PA1_rVAR) of the anteroposterior ankle sway movement component than balancing with open eyes, indicating faster anteroposterior ankle sways during closed-eye balancing. This suggests that closing eyes during balancing results in faster postural sway and decreased stability control of the first main movement component, especially on unstable surfaces, compared to balancing with open eyes. The findings support the hypothesis that balancing with the closed eye condition makes it more challenging to control postural stability than balancing with open eyes, mainly when the balance tasks are performed on unstable surfaces. Additionally, a high magnitude of neuromuscular control (PA1_RMS) was observed in unstable conditions for the closed eye condition.

Point 8: L312-13 The author indicates that the findings suggest that the inherent age and visual contribution effects on postural control should be considered for training, injury prevention, and rehabilitation. Same, what's new here?

Response 8: The new findings suggest that the inherent age and visual contribution effects on postural control should be considered for training, injury prevention, and rehabilitation. Older adults and those balancing with closed eyes exhibit inferior neuromuscular performance caused by inherent degenerative changes in the neuromuscular system and by omitting visual input, respectively. Specifically, the dorsi flexor and plantar flexor muscles play an essential role in the anteroposterior ankle strategy, and their age-related degenerations have been reported as a possible risk of falls. To prevent injuries and promote recovery from injury, older adults may benefit from specific training programs that focus on enhancing neuromuscular control of the lower extremities and other body parts, such as core and upper limb muscles, on slowing aging effects on postural control. Similarly, individuals who have experienced an injury affecting postural stability may need tailored rehabilitation protocols that target neuromuscular performance, including balancing training on unstable surfaces and omitting visual feedback to improve their ability to maintain postural stability.

Reviewer 5 Report

The author presents an interesting and original paper investigating the effects of age and vision and their relative contribution on postural control.

This is an area that has received an extensive attention in the literature, therefore, warrants further examination. Overall, the manuscript is written and organized fairly well. It follows the logical sequence of a research purpose. Despite this strength, I have a major comment that need to be addressed by the author and listed below.

INTRODUCTION
The introduction is clear and present key papers related to the aim of the study. However and related to my major concern (see Materiels and Method section) maybe a part of the literature concerning the postural control determinants is missing (and adjusted too in the discussion if needed and relevant).

MATERIAL and METHODS

According to the author, subjects with higher BMIs have been excluded. Since it has been clearly shown that obesity affects postural control, I agree with the author for this decision. However, other anthropometric parameters have been identified to influence postural control (body height, body weight/BMI and not only obesity itself; see for example Chiari L, Rocchi L, Cappello A. Stabilometric parameters are affected by anthropometry and foot placement. Clin Biomech (Bristol, Avon). 2002; 17(9-10):666-77; Corbeil P, Simoneau M, Rancourt D, Tremblay A, Teasdale N. Increased risk for falling associated with obesity: mathematical modeling of postural control. IEEE Trans Neural Syst Rehabil Eng. 2001; (2):126-36; Hue O, Simoneau M, Marcotte J, Berrigan F, Doré J, Marceau P, Marceau S, Tremblay A, Teasdale N. Body weight is a strong predictor of postural stability. Gait Posture. 2007;26(1):32-8). Or there is a significant difference between groups in the current study concerning these variables. I would suggest to the author to check his analyses by adjusting the influence of these variables (ANCOVAs) to confirm his current results. Indeed, a question still remains:  is it only an age/vision effect  observed or a bias effect related to the contribution of anthropometric variables or maybe a combined effect?

RESULTS and CONCLUSION

Related to my previous comment, these sections might need to be changed/modified.

Author Response

Response to Reviewer 5 Comments

Point 1: The author presents an interesting and original paper investigating the effects of age and vision and their relative contribution on postural control.

This is an area that has received an extensive attention in the literature, therefore, warrants further examination. Overall, the manuscript is written and organized fairly well. It follows the logical sequence of a research purpose. Despite this strength, I have a major comment that needs to be addressed by the author and listed below.

INTRODUCTION

The introduction is clear and present key papers related to the aim of the study. However and related to my major concern (see Materiels and Method section) maybe a part of the literature concerning the postural control determinants is missing (and adjusted too in the discussion if needed and relevant).

MATERIAL and METHODS

According to the author, subjects with higher BMIs have been excluded. Since it has been clearly shown that obesity affects postural control, I agree with the author for this decision.

However, other anthropometric parameters have been identified to influence postural control (body height, body weight/BMI and not only obesity itself; see for example Chiari L, Rocchi L, Cappello A. Stabilometric parameters are affected by anthropometry and foot placement. Clin Biomech (Bristol, Avon). 2002; 17(9-10):666-77; Corbeil P, Simoneau M, Rancourt D, Tremblay A, Teasdale N.

Increased risk for falling associated with obesity: mathematical modeling of postural control. IEEE Trans Neural Syst Rehabil Eng. 2001; (2):126-36; Hue O, Simoneau M, Marcotte J, Berrigan F, Doré J, Marceau P, Marceau S, Tremblay A, Teasdale N. Body weight is a strong predictor of postural stability. Gait Posture. 2007;26(1):32-8).

Or there is a significant difference between groups in the current study concerning these variables. I would suggest to the author to check his analyses by adjusting the influence of these variables (ANCOVAs) to confirm his current results.

Indeed, a question still remains:  is it only an age/vision effect  observed or a bias effect related to the contribution of anthropometric variables or maybe a combined effect?

RESULTS and CONCLUSION

Related to my previous comment, these sections might need to be changed/modified.

Response 1: Thank you for taking the time to review my article. I appreciate your insightful comments and suggestions. Your feedback has been precious in improving the quality of the work. I have carefully considered your suggestions and made the necessary revisions to address your issues.

Could I explain or inform you about the methodology used in the current study, especially the pre-processing procedure used to prepare the matrix for PCA analysis?

According to your comments, our laboratory, which originates the PManalyzer software used to extract movement components/synergies, is concerned about the effects of anthropometric differences between the participants, so we have a solution to address this point by having three data pre-processing procedure (see mathematical details in Haid et al. 2019 doi:10.3389/fninf.2019.00024 or in Promsri et al. 2020 10.3390/brainsci10030128) to solve the problem of anthropometric differences.

Hence, I apologize for not running the new analysis based on your comments. In addition, I slightly rewrote the pre-processing procedures to make them more straightforward.

Once again, I sincerely appreciate your time and effort in reviewing our article.

Round 2

Reviewer 4 Report

Revision is adequate.

Reviewer 5 Report

Thank you for all the modifications made. Your manuscript, especially concerning the used method, is more clear and detailed and futher analyses, as previously suggested/required, are not relevant after your explanations and references added.